# An Investigation of the Etiologies of Non-Immune Hydrops Fetalis in the Era of Next-Generation Sequence—A Single Center Experience

**DOI:** 10.3390/genes13122231

**Published:** 2022-11-28

**Authors:** Xing Wei, Yingjun Yang, Jia Zhou, Xinyao Zhou, Shiyi Xiong, Jianping Chen, Fenhe Zhou, Gang Zou, Luming Sun

**Affiliations:** Department of Fetal Medicine and Prenatal Diagnosis Center, Shanghai First Maternity and Infant Hospital, Tongji University School of Medicine, Shanghai 201204, China

**Keywords:** non-immune hydrops fetalis, causes, etiologies, genetic testing, exome sequencing

## Abstract

(1) Background: Numerous etiologies may lead to non-immune hydrops fetalis (NIHF). However, the causes remain unclear in half of NIHF cases following current standard assessment. The application of prenatal chromosomal microarray analysis (CMA) and exome sequencing (ES) can improve the identification of the etiologies. This study aimed to investigate the etiologies of NIHF in the era of next-generation sequence (NGS) following a unified prenatal work-up flow for diagnosis. (2) Methods: A retrospective analysis was conducted on NIHF cases that were collected prospectively to explore the underlying etiologies according to a unified prenatal diagnosis work-up flow at Shanghai First Maternity and Infant Hospital between Jan 2016 and Dec 2019. The medical records for all NIHF cases were reviewed, and the causes of NIHF were classified as confirmed (diagnostic), suspected, or unknown. (3) Results: Prenatal and postnatal medical records for a total of 145 NIHF cases were reviewed, 48.3% (70/145) of the cases were identified to be with confirmed etiologies, and 10.3% (15/145) with suspected etiologies. Among 85 cases with confirmed or suspected etiologies, 44.7% were diagnosed with genetic disorders, 20% with chylothorax/chyloascites diagnosed postnatally, 12.9% with fetal structural anomalies, 12.9% with fetal anemia, 7% (6 cases) with fetal arrhythmia, and 2.3% (2 cases) with placenta chorioangioma. In cases with genetic disorders, 8 aneuploidies were detected by CMA, and 30 cases had single-gene disorders identified by ES (29/30) or targeted gene panel (1/30). There were still 41.4% cases (60/145) with unknown causes after this unified prenatal diagnostic work-up flow. (4) Conclusions: In the era of NGS, the causes of NIHF were identified in 58.6% of cases, with genetic disorders being the most common ones. NGS is helpful in determining the genetic etiology of NIHF when CMA results cannot explain NIHF, but 41.4% of cases were still with unknown causes under the unified prenatal diagnostic work-up flow in this single-center study.

## 1. Introduction

Hydrops fetalis occurs in approximately 1 in 1700 to 3000 pregnancies and is diagnosed by prenatal ultrasound when at least two abnormal fetal fluid collections are present [1]. With the administration of Rh (D) immune globulin and intrauterine blood transfusion for severe fetal anemia screened by middle cerebral artery peak systolic velocity (MCA-PSV), fetal hydrops caused by alloimmunization (immune hydrops fetalis, IHF) has dramatically decreased. Non-immune hydrops fetalis (NIHF), defined as fetal hydrops after maternal red-cell alloimmunization was ruled out, now accounts for almost 90% of hydropic cases [1,2]. The etiology of NIHF is complex and diverse and associated with various adverse perinatal outcomes [3,4,5]. It is crucial for a fetal medicine specialist to explore the underlying etiologies of NIHF to help provide appropriate prenatal counseling and decision making.

A standard work-up for the evaluation of NIHF etiologies prenatally was recommend by the Society for Maternal-Fetal Medicine (SMFM) in 2015 [6]. Following the SMFM guidelines, confirmed etiologies were found in 44% (29/65) of NIHF cases by a retrospective study conducted in five fetal medicine centers. The study’s most common etiologies were aneuploidies and cardiovascular anomalies when exome sequence was not widely applied prenatally [7]. The author provided targeted gene panels for 9 cases and ES for 2 highly selected cases and identified 4 cases (36%) with single-gene disorders. With the development of molecular genetics, several studies focusing on the genetic etiologies of prenatally diagnosed NIHF explored the value of NGS in NIHF cohorts and found that the detection rates of CMA and ES were 0~4.2% and 29~50%, respectively [8,9,10,11,12,13]. Furthermore, Sparks and colleagues also compared the diagnostic yield of ES with the target gene panel in a NIHF cohort and found that the ES was superior [14]. However, few studies have explored the overall etiology distribution of NIHF in the era of NGS.

Therefore, we aimed to assess the proportion and distribution of confirmed etiologies in strictly defined NIHF cases in a fetal treatment center that provides CMA and ES for all cases and to explore the value of prenatal CMA and ES for aiding clinical counseling.

## 2. Materials and Methods

### 2.1. Study Design and Participants

This was a retrospective analysis of 145 single NIHF cases prospectively collected that were referred to the Department of Fetal Medicine & Prenatal Diagnosis Center at Shanghai First Maternity and Infant Hospital of Tongji University between January 2016 and December 2019. Prenatal diagnostic criteria for fetal hydrops are at least two abnormal fluid collections in fetal cavities or soft tissue, including pleural effusion, ascites, pericardial effusion, or skin edema. After ruling out maternal red-cell alloimmunization, NIHF was diagnosed.

The study was approved by the ethics committee of Tongji University (registration number: 2018yxy13), and informed consent was obtained from the patients.

### 2.2. Procedure

Data were prospectively collected from the medical record, including maternal characteristics, previous adverse pregnancy history (fetal hydrops, fetal death, or neonatal death), gestational age at diagnosis, ultrasound manifestations, and pregnancy outcomes. Chinese NIHF guidelines refer to the SMFM and SOGC NIHF guidelines and make some necessary modifications [6,15]. According to Chinese guidelines, antenatal investigations of NIHF include (Figure 1): (1) maternal blood group and erythrocyte antibody screen, blood counts and hemoglobin electrophoresis if necessary; TORCH serology (toxoplasma, rubella, cytomegalovirus, syphilis, and herpes). When fetal MCA-PSV was elevated, we provided parvovirus screening and Kleihauer–Betke (K-B) test; (2) detailed fetal anatomy scan, echocardiography, Doppler evaluation for MCA-PSV; and (3) related genetic testing: including CMA, karyotyping, targeted gene panel, and prenatal or postnatal ES if the results of CMA could not explain NIHF.

The sources of DNA samples for genetic testing included prenatal samples from chorionic villi, amniotic fluid, or umbilical cord blood or postnatal samples from fetal tissues. We provided CMA with or without karyotyping for genetic assessment as the first step. For cases without diagnostic disorders for NIHF, i.e., CMA results were negative or CNVs detected by CMA could not explain fetal hydrops, namely unexplained NIHF, ES was provided for further evaluation.

SNP array analysis was performed using CytoScan 750 k (Affymetrix Inc., Santa Clara, CA, USA). The copy number variations (CNVs) were shown according to the human Feb. 2009 (GRCh37/hg19) assembly. ES was performed using the Agilent SureSelect v6 (cat No.5190–8864, Agilent, Santa Clara, CA, USA) kit and sequenced with an Illumina Hiseq X Ten (Illumina, San Diego, CA, USA). Variants were annotated and filtered by Ingenuity Variant Analysis (Ingenuity Systems, Redwood City, CA, USA). Common variants were filtered by the databases of the Exome Aggregation Consortium (ExAC) (http://exac.broadinstitute.org, accessed on 10 September 2022), the Exome Sequencing Project (https://esp.gs.washington.edu, accessed on 10 September 2022), the 1000 Genomes Project (http://www.1000genomes.org, accessed on 10 September 2022), and an internal database. The remaining phenotype-related variants were then assessed under the protocol issued by the American College of Medical Genetics and Genomics/Association for Molecular Pathology (ACMG/AMP) guidelines [16]. All putative variants detected by WES were confirmed by polymerase chain reaction (PCR) and Sanger sequencing in each fetus and their parents for testing the origin and phase of the variants.

The etiologies were categorized into three groups: confirmed, suspected, or unknown through prenatal and postnatal evaluations. Confirmed etiologies were those that were strongly supported by the existing literature as leading to NIHF, including (1) fetal structural anomalies, including congenital pulmonary airway malformation (CPAM) and Galen aneurysm without genetic anomalies, fetal arrhythmia, and placenta chorioangioma; (2) genetic disorders, including aneuploidies, pathogenic CNVs, and single-gene disorders; (3) fetal anemia that was not associated with genetic disorders and confirmed by FBS; (4) etiologies confirmed by postnatal investigations, including chylothorax, chyloascites, and meconium peritonitis. In the present study, cases with genetic causes that were accompanied by abnormal ultrasonic phenotype were classified as genetic anomalies.

Suspected etiologies were those that may explain NIHF, including: variants with unknown significance (VUS) of already known genes detected by ES; fetal anemia confirmed by FBS; fetal structural anomalies such as right cardiac dysplasia syndrome, and multiple structural abnormalities that may cause NIHF, but the genetic assessment was not performed due to insufficient DNA samples.

The rest were categorized as cases with unknown causes.

### 2.3. Outcomes

The primary outcomes were the proportion of cases with clearly confirmed etiologies and their distribution. The secondary outcomes were the detection rate of pathogenic (P) and likely pathogenic (LP) variants that can explain NIHF by CMA and WES.

### 2.4. Statistical Analysis

SAS 9.3 software was used to perform data analysis. For continuous variables, mean and standard deviation (SD) or median (range) were adopted according to their distribution. For categorical variables, number and proportion/rate were used to describe their distribution.

## 3. Results

### 3.1. Study Participants

A total of 145 NIHF cases were included during the study period. The demographics and clinical characteristics of the cohort are presented in Table 1. The median gestational age at diagnosis was 24.3 weeks (range, 13~35), 21.4% were recurrent NIHF, and 39.3% were accompanied by abnormal structures. From the patient group, 102 patients (70.3%) chose to terminate their pregnancy for fear of the poor prognosis. Of the 43 cases that continued the pregnancy, the incidence of intrauterine fetal death was 11.6% (5/43), and the neonatal survival rate was 63.2% (24/38).

Figure 2 shows the genetic assessments of 145 NIHF cases, among which one underwent thalassemia-targeted gene testing since the couple were both carriers of α thalassemia minor (--/αα), and 144 cases did CMA. ES was performed in 71% of unexplained cases (96/136). For the sources of fetal DNA, 76.6% (111 cases) were from amniotic fluid, 10.3% (15 cases) from fetal blood, 10.3% (15 cases) from fetal tissues after termination, and 2.8% (4 cases) from chorionic villus.

### 3.2. The Overall Etiologies in the NIHF Cohort

Overall, 48.3% (70/145) of the cases were identified to be with confirmed (diagnostic) etiologies, 10.3% (15/145) with suspected etiologies, and 41.4% (60/145) with unknown etiologies. Among the 85 cases with confirmed or suspected etiologies, 44.7% (38/85) were diagnosed to be with genetic disorders, including 8 aneuploidies and 30 single-gene diseases, 20% (17/85) with postnatally diagnosed chylothorax/chyloascites, 12.9% (11/85) with fetal anemia, 12.9% (11/85) with fetal structural anomalies, 7% (6/85) with fetal arrhythmia, and 2.3% (2/85) with placenta chorioangioma (Figure 3).

A total of 70 cases were diagnosed with confirmed etiologies, of which 45.7% (32/70) were genetic disorders including 8 aneuploidies and 24 single gene disease of which 23 were detected with ES; 1 case was diagnosed with Bart’s hydrops fetalis using a thalassemia gene panel, 24.3% (17/70) with postnatally diagnosed chylothorax/chyloascites, 11.4% (8/70) with explained fetal anemia, 7.1% (5/70) with fetal structural anomalies; 6 cases were fetal arrhythmia, and 2 were placenta chorioangioma. Among the 15 cases with suspected causes, the etiologies in overall descending order of frequency were ultrasound anomalies (6 cases, 40%), single-gene disorders (6 cases, 40%), and fetal anemia (3 cases, 20%). Of the 60 cases with unknown etiology, 68.3% (41 cases) completed ES assessment, but the cause was still unknown.

### 3.3. The Detection Yield of CMA

The detection rate of CMA was 7.6% (11/144). Among them, 8 cases (5.6%) were aneuploidies, including 6 cases with trisomy 21, 1 with Turner syndrome, and 1 with trisomy 13. In addition to aneuploidies, CMA identified 3 cases (2.1%) with P/LP CNVs but could not explain NIHF.

### 3.4. The Detection Yield of ES

Exome sequencing was performed in 96 cases, of which 24 cases received ES prenatally, and 72 cases did so postnatally using stored fetal DNA samples. P/LP gene variants were detected in 23.9% (23 cases), among which 8 were recurrent hydrops (2 cases with GUSB and 6 with 1 of GBA, LZTR1, FOXC2, FOXP3, GBE1, or RAPSN); 15 were non-recurrent NIHF (4 cases with PTPN11 gene, 11 with 1 of MAP2K2, ALG12, CDAN1, LZTR1, ANKRD11, KRAS, NRAS, EPHB4, PIK3CA, NEB, or GUSB). Six cases of recurrent NIHF had recessive inheritance patterns, of which four were lysosomal storage disorders. Autosomal dominant inheritance accounted for 73.3% (11/15) of non-recurrent NIHF, and Noonan syndrome was the most common one (63.6%, 7/11).

In addition to LP or P variants, 6 (6.3%) cases were found to have gene variants with potential clinical significance but did not meet the criteria for P or LP and were considered suspected causes. Among the six cases, RAPSN gene variants were found in three cases with recurrent NIHF, PROC gene variants in one recurrent NIHF case, and PIEZO1 gene variants were found in two patients (one was recurrent NIHF).

After careful evaluation, among 67 cases with normal WES results, 23 cases were identified with confirmed etiologies, including 8 cases of fetal anemia, 13 cases of postpartum chylothorax/chyloascites, 1 case of Galen aneurysm, and 1 case of SVT. There were three cases with suspected etiologies, including one with multiple structural abnormalities, one with right ventricular dysplasia, and one with intracranial structural abnormalities. Even after the completion of ES, there were still 41 cases with unknown etiology.

### 3.5. The Etiologies of Cases That Did Not Perform ES

Among 40 cases that did not perform ES, 15 had confirmed etiologies (11 cases with ultrasound anomalies and 4 with postnatally diagnosed chylothorax). There were three cases with suspected fetal anemia diagnosed by FBS and three cases with suspected ultrasound anomalies (one with microcephaly accompanied with right ventricular hypoplasia, one with umbilical–portal–systemic venous shunt, and one with multiple structural abnormalities). Still, ES was not performed due to insufficient DNA sample. The remaining 19 cases were assigned to the unknown etiology group, where no ultrasound abnormality could explain NIHF, and their DNA samples were insufficient for ES.

## 4. Discussion

This prospective study showed the distribution of the etiologies in NIHF, of which 48.3% were identified with clearly confirmed etiologies, while 10.3% had suspected etiologies, and 41.4% remained unknown etiologies. Among the confirmed and suspected etiologies, the frequency in descending order were genetic diseases, chylothorax/chyloascites, fetal structural anomalies, and fetal anemia. CMA did not yield any pathogenic CNVs related to NIHF beyond aneuploidies. In NIHF cases with unexplained CMA results, ES could identify P or LP gene variations in 23.9% of the cases and potentially diagnostic variants in an additional 6.3% of the cases.

In this strictly defined NIHF cohort study, the percentage of NIHF with confirmed etiology was similar to Sparks’s (48% vs. 44%) [7] but was smaller than other published series. Previous studies included all fetal structural anomalies that might cause NIHF, such as congenital cardiac defects or multiple structural abnormalities that might be the prenatal phenotype of genetic disorders, and categorized them as confirmed etiologies associated with NIHF [8,17,18,19]. In theory, the proportion of confirmed cases should be higher than Sparks’s, as 66% of cases performed ES in the present study. This may be because our center is a fetal therapy referral center, the majority of cases chose termination (70.3% vs. 20% in Sparks’s) after counseling, resulting in difficulties in obtaining the postpartum diagnosis, such as chylothorax, which was rarely associated with genetic disorders.

Genetic disorder (26%) was the most common cause of NIHF in this study. Different from SMFM guidelines for NIHF, due to the high incidence of thalassemia in southern China, thalassemia screening is included in the Chinese guidelines as a routine procedure. Since the diversity of Shanghai’s population, only one case was diagnosed with Bart’s hydropic fetalis. CMA did not identify any pathogenic CNVs related to NIHF beyond aneuploidies, which was consistent with previous studies [7,8]. The proportion of chromosomal abnormalities in this study was much lower than Khalil’s [19] (5.5% vs. 28.4%). It was due to the fact that in Khalil’s study, far more cases were diagnosed before 14 weeks (26% vs. 4.8%), which was obviously associated with a higher prevalence of chromosomal abnormalities. In strictly defined prenatal NIHF cases that received WES, causative gene variants were found in 23.9%. It was slightly lower than the study conducted by Sparks (29%) [11], who enrolled cases with increased thickness of nuchal translucency, cystic hygroma, and abnormal fluid collection in a single body cavity. However, consistent with Sparks’s findings, in NIHF cases with single gene diseases, RASopathies (39%) and metabolic disorders (26%) were the most common. While the proportion of autosomal recessive disorders in this study was higher than that in Sparks (39.1% vs. 27%), it was related to the fact that 21.4% of recurrent NIHF cases were included in this study, and such cases tend to have a higher incidence of recessive inheritance [13].

The incidence of confirmed ultrasound abnormalities that could explain NIHF was also smaller than in Khalil’s study (8.9% vs. 24%), which may be due to the different definitions of confirmed ultrasound abnormalities. We only defined ultrasound anomalies rarely associated with genetic variations as confirmed ultrasound anomalies, including congenital pulmonary airway malformation, fetal arrhythmias, and placental chorioangioma [20,21,22], similar to Sparks’s study where the proportion of cases with ultrasound anomalies was 13.4% [7]. Different from other published series, NIHF cases with severe pleural effusion were referred to our fetal therapy center for intrauterine intervention. Therefore, the proportion of NIHF cases diagnosed with chylothorax/chyloascites (11.7%) was higher. NIHF caused by unexplained fetal anemia accounted for 5.5% (8 cases), among which 3 were recurrent fetal anemia with negative ES results and treated by intrauterine transfusion with good perinatal outcomes.

There are several strengths of this study. It was a large cohort of prenatally diagnosed singleton NIHF cases focused on the distribution of genetic and non-genetic etiologies of NIHF, which was helpful for clinical counseling and the management of NIHF in a fetal therapy center. We had a relatively large sample size that performed exome sequencing, which made our results more robust and reliable. Furthermore, the present study included only singleton NIHF cases in which the etiologies would be more easily clarified.

However, this study was not without limitations. This was a single-center prospective cohort study, and the etiologies’ distribution could not represent the complete profile of all NIHF. Cases with confirmed aneuploidies that were identified in other institutions might terminate the pregnancy before referral. In addition, about 70% of cases chose termination of the pregnancy after evaluation due to concerns about poor prognosis, the information of their postpartum diagnosis, and their natural course was unable to obtain.

## 5. Conclusions

In conclusion, 58% of the prenatal NIHF cases could be identified with etiologies, while 42% remained with unknown etiologies. The frequency of etiologies by descending order were genetic disorders, chylothorax/chyloascites, fetal structural anomalies, and fetal anemia. CMA did not yield the detection of pathogenic CNVs, while ES improved the identification of genetic disorders with a detection rate of 24% in explained NIHF. Whole-genome sequencing or other genetic tests may be recommended to identify the etiology in NIHF cases with negative ES results.

## Figures and Tables

**Figure 1 genes-13-02231-f001:**
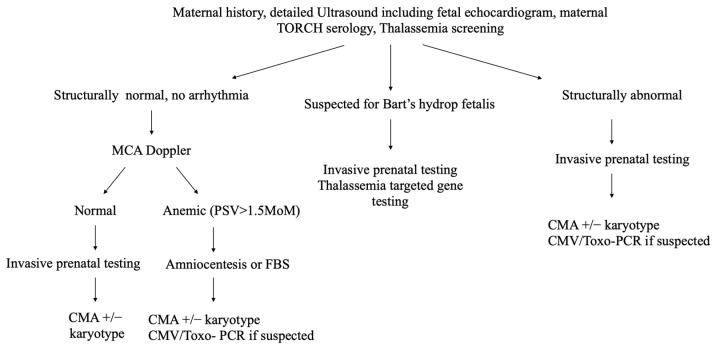
Prenatal work-up of non-immune hydrops fetalis; FBS: fetal blood sampling; CMV: cytomegalovirus.

**Figure 2 genes-13-02231-f002:**
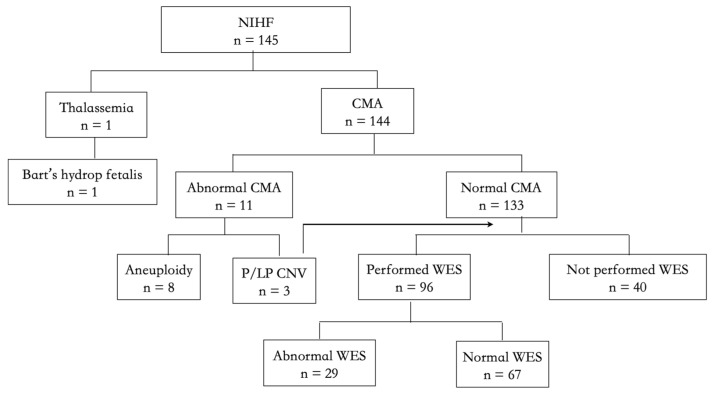
Flow chart of genetic assessment of the NIHF cohort.

**Figure 3 genes-13-02231-f003:**
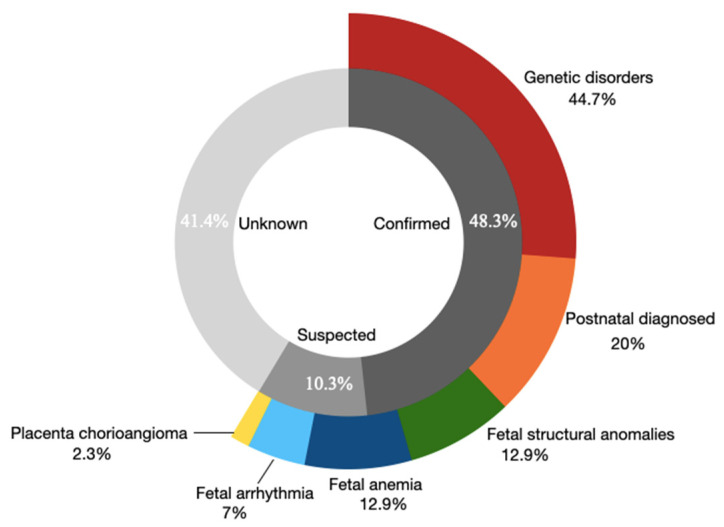
The distribution of etiologies in NIHF cohort.

**Table 1 genes-13-02231-t001:** Clinical characteristics of 145 NIHF cases.

Characteristic	Values
Maternal age, years, mean ± SD	29.6 ± 4.6
Spontaneous conception, n (%)	132 (91.0)
Recurrent NIHF ^§^, n (%)	31 (21.4)
GA at diagnosis, wks, median (range)	24.3 (13.0, 35.0)
<16 weeks, n (%)	15 (10.3)
Accompanied with abnormal structure ^‡^, n (%)	57 (39.3%)
Termination of pregnancy, n (%)	102 (70.3%)
Continued pregnancy, n (%)	43 (29.7%)
Intrauterine fetal death, n (%)	5 (11.6%)
Neonatal death, n (%)	14 (36.8%)

^§^ Having a previous pregnancy of non-immune fetal hydrops. ^ǂ^ Including cases with ultrasound soft markers.

## Data Availability

The data presented in this study are available on request from the corresponding author. The data are not publicly available due to the privacy of the patients.

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
