# Peer review of "An Investigation of the Etiologies of Non-Immune Hydrops Fetalis in the Era of Next-Generation Sequence—A Single Center Experience"

_genes, 2022, doi:10.3390/genes13122231_

Round 1
Reviewer 1 Report
The paper by Xing et al. is a nice retrospective analysis of individuals with non-immune hydrops fetalis from a large fetal medicine and prenatal diagnosis center. The study is well described and clearly details how the individual samples were tested and what the findings were from the cohort. The findings in this paper do not match findings from other similar studies and the authors do a nice job detailing why they think the results might differ. I found the explanations to be helpful in understanding the complexities of these types of studies. I do not have any strong suggestions for the authors. I did note that the thalassemia case may have been left out of the explanation of etiologies in section 3.2. It is noted that there are 24 single gene disorders and 23 are explained under ES. I may have missed the mention of the thalassemia case but I think it may be the 24th case and may have not been mentioned in this section (not an ES case so not sure where it belongs).
Author Response
Response: Thank you very much for your kind comments. ES yield out 23 cases with single gene disorders. The 24th case is a Bart’s hydropic fetalis not detected by ES but by thalathemia gene panel. Since the case of Bart’s hydropic fetalis belongs to the single gene disorder, in section 3.2, we included the cases in the 24 single gene diseases. Sorry, we forgot to provide the details of Bart’s case. We have added this information to the manuscript.
Reviewer 2 Report
In this paper, the authors used chromosomal microarray analysis and exome sequencing to identify the etiologies of the non-immune hydrops fetalis (NIHF). The genetic disorders were identified as the most common etiology of NIHF. However, there were still 41.4%cases with unknown causes. Overall, it is an interesting paper, but the authors should detail their results and discuss them more sufficiently. Some points to consider:
1. The authors used chromosomal microarray analysis and exome sequencing to identify the etiologies of NIHF. However, we can only see the description of the microarray and sequencing results. We don’t get many details, like the pathway analysis, GO term, KEGG, et al. They should provide more figures or tables to introduce their results. We should get the most important conclusion from their figures or tables. From their current results, it is hard to get their crucial information.
2. The most common etiologies of NIHF identified in this study are genetic disorders. The authors should detail these disorders and discuss the potential mechanisms.
3. Most of their findings have been introduced in previous studies, and we can not get a valuable conclusion from their current results. It is hard to know how to explore the value of prenatal CMA and ES to aid clinical counseling from their current workflow.
4. They should provide an online source to show the original microarray and sequencing data.
Author Response
1. The authors used chromosomal microarray analysis and exome sequencing to identify the etiologies of NIHF. However, we can only see the description of the microarray and sequencing results. We don’t get many details, like the pathway analysis, GO term, KEGG, et al. They should provide more figures or tables to introduce their results. We should get the most important conclusion from their figures or tables. From their current results, it is hard to get their crucial information.
Response 1: Thanks for your constructive suggestions. In this study, we mainly aimed to provide evidence to help clinical consultation on the etiologies for NIHF, including genetic and non-genetic factors, and further guide clinical management. Therefore we did not include details about variants related to bioinformatic analysis of detected gene disorders in this study. We are willing to provide detailed information on the genes with P/LP variants detected by ES in supplementary tables if you request.
2. The most common etiologies of NIHF identified in this study are genetic disorders. The authors should detail these disorders and discuss the potential mechanisms.
Response 2: Thank you for the comment. This study is a clinical study related to the etiology distribution of NIHF, we aimed to provide some data to help clinical counseling and management. We will describe the details of the genes involved and the underlying mechanisms in subsequent papers focusing on the genetic etiology of NIHF.
3. Most of their findings have been introduced in previous studies, and we can not get a valuable conclusion from their current results. It is hard to know how to explore the value of prenatal CMA and ES to aid clinical counseling from their current workflow.
Response 3: The detection rate of ES was similar to Spark’s study that was published in NEJM. Spark’s study included cases with increased NT and single cavity edema, while the present study enrolled strictly defined NIHF cases with a relatively large sample, and the detection rate of ES (abnormal ES result, including P/LP and VOUS) in NIHF cases with negative CMA results was about 30%.
4. They should provide an online source to show the original microarray and sequencing data.
Response 4: We thank the reviewer for the comment. We did not provide the online source for the original due to the following reasons: 1) the protection of the privacy of the patients and genetic sources; 2) further research will be conducted on this data.
Round 2
Reviewer 2 Report
In their revised version, the authors have addressed my concerns. This manuscript has my support for publication in Genes without further edits or experimentation.